# Microalgae Bioactive Compounds to Topical Applications Products—A Review

**DOI:** 10.3390/molecules27113512

**Published:** 2022-05-30

**Authors:** Manuel Martínez-Ruiz, Carlos Alberto Martínez-González, Dong-Hyun Kim, Berenice Santiesteban-Romero, Humberto Reyes-Pardo, Karen Rocio Villaseñor-Zepeda, Edgar Ricardo Meléndez-Sánchez, Diana Ramírez-Gamboa, Ana Laura Díaz-Zamorano, Juan Eduardo Sosa-Hernández, Karina G. Coronado-Apodaca, Ana María Gámez-Méndez, Hafiz M. N. Iqbal, Roberto Parra-Saldivar

**Affiliations:** 1School of Engineering and Sciences, Tecnológico de Monterrey, Monterrey 64849, Mexico; mmruiz@tec.mx (M.M.-R.); a01283495@itesm.mx (C.A.M.-G.); a01252545@itesm.mx (D.-H.K.); a01113721@itesm.mx (B.S.-R.); hreyesp@tec.mx (H.R.-P.); rocio.vz@tec.mx (K.R.V.-Z.); edgar.rmelendez@tec.mx (E.R.M.-S.); diana.ramirez.gamboa@tec.mx (D.R.-G.); ana_diaz@tec.mx (A.L.D.-Z.); eduaro.sosa@tec.mx (J.E.S.-H.); karina.coronado@tec.mx (K.G.C.-A.); 2Basic Sciences Department, Universidad de Monterrey, San Pedro Garza García 66238, Mexico; ana.gamezm@udem.edu

**Keywords:** microalgae, bioactive compounds, photoprotectants, immunomodulator, antioxidants, biomass

## Abstract

Microalgae are complex photosynthetic organisms found in marine and freshwater environments that produce valuable metabolites. Microalgae-derived metabolites have gained remarkable attention in different industrial biotechnological processes and pharmaceutical and cosmetic industries due to their multiple properties, including antioxidant, anti-aging, anti-cancer, phycoimmunomodulatory, anti-inflammatory, and antimicrobial activities. These properties are recognized as promising components for state-of-the-art cosmetics and cosmeceutical formulations. Efforts are being made to develop natural, non-toxic, and environmentally friendly products that replace synthetic products. This review summarizes some potential cosmeceutical applications of microalgae-derived biomolecules, their mechanisms of action, and extraction methods.

## 1. Introduction

Cosmetics are products designed to improve the appearance of the skin without affecting its function and structure due to the active ingredients contained in the product [1,2]. On the other hand, the cosmetic industry uses the word “cosmeceuticals” to refer to cosmetic products with ingredients that have medicinal or drug-like benefits [1,3]. The vast majority of cosmeceutical products are lotions or creams for topical use. Although the terms may seem confusing, and cosmeceuticals are not formally recognized by the United States Food and Drug Administration (US-FDA) or the European Union, dermatologists prescribe 30–40% of these products [4]. Furthermore, the global natural cosmetics market showed 10–11% annual growth from 2015 to 2019 [5]. While botanical cosmeceuticals are in demand and display numerous benefits, the potential of microalgae as an alternative source for cosmeceutical products has been extensively studied [6,7,8]. 

Algae are a diverse group of photosynthetic eukaryotic microorganisms that possess different structures and forms and can be divided into either macro or microalgae based on their size. Many microalgae species have been identified as having various biochemical characteristics associated with nutritional benefits and human health. 

Microalgae synthesize an extensive diversity of compounds from different metabolic pathways such as amino acids, fatty acids, lycopene, polysaccharides, steroids, carotenoids, lectins, polyketides, toxins, etc. Some of these are shown in Figure 1.

Microalgae produce and secrete valuable metabolites as a result of being constantly exposed to several stress conditions, such as high or low temperature, high salinity, osmotic pressure, photo-oxidation, and ultraviolet radiation [9]. Within the group of organisms with photosynthetic capacity are microalgae and cyanobacteria, microorganisms classified as eukaryotes and prokaryotes, respectively [10]. Both present great biodiversity of species, ranging between 80,000 and 200,000 [11]. Microalgae and cyanobacteria are responsible for 32% of photosynthetic activity in the world; therefore, they play an important role in various ecosystems [12]. The biotechnological interest in microalgae and cyanobacteria has focused on their content of valuable metabolites that can be utilized to develop a wide range of different applications in fields such as medicine, industry, energy, agriculture, food, and others. Stress factors increase or decrease the production of different convenient compounds; for this reason, scientific research focuses on defining the best microalgae cultivation conditions to obtain the maximum industrial production [13]. Furthermore, it is widely recognized that microalgae metabolites exhibit a wide range of biological activities such as ultraviolet (UV) absorbing, antioxidant, anti-aging, anti-blemish, anti-inflammatory, and antimicrobial properties [1,14]. Additionally, microalgae capture CO_2_ more efficiently than trees, reducing the greenhouse effect [15].

Some of the advantages of microalgae include a higher annual photon-to-biomass conversion efficiency in comparison with plants, as reported in previous studies; microalgae can growth in wastewater as treatment for water bioremediation [16,17,18,19,20,21]. Even though it is possible to use wastewater to produce microalgal biomass and obtain bioactive compounds, their use for topical applications or human consumption still represents a great challenge due to the diversity of contaminants present in wastewater, such as pathogens and heavy metals, despite pre-treatment steps such as sterilization or filtration to remove bacteria and fungi and recovery of heavy metals using coagulation–flocculation methods [22]. Pre-treatment also helps to reduce the concentration of nutrients and the turbidity of the water to allow the penetration of light necessary for microalgal growth [23,24]. Furthermore, methods of extraction and purification of compounds of interest are being investigated for their safe application in cosmetic and nutraceutical products [25,26].

Currently, some microalgae-derived products are available on the market, and the biotechnology of microalgae is attracting more attention. Labels such as ©DSM use *Nannochloropsis oculata* and *Dunaliella salina* extract in some of their products, such as PEPHA^®^-TIGHT CB or PEPHA^®^-CTIVE CB, to provide skin-tightening effects and stimulate cell proliferation, respectively [27]. However, an ever-increasing effort is being invested in the search for new environmentally friendly cosmetic products. All of the diverse and interesting features qualify microalgae metabolites as an alternative feedstock or potential source of biomolecules of commercial interest. 

The present review describes the accumulated knowledge about the foremost groups of cosmeceutical compounds from microalgae metabolites and putative drug delivery methods using microalgae-based biotechnology.

## 2. Immunomodulatory Activity of Compounds from Microalgae

Microalgae and cyanobacteria serve as a source of bioactive compounds that present immunomodulatory activity, such as amino acids [28,29,30], peptides, pigment-protein complex, and exopolysaccharides, the latter known to possess anti-inflammatory, antimicrobial, and antiviral properties [1,31,32,33,34]. The major sugars involved in the composition of polysaccharides are xylose, glucose, mannose, and galactose [35]. Specifically, sulfated polysaccharides have been associated with blood coagulation, antilipidemic activity, and mainly immunomodulatory activity [31]. Multiple studies have reported anti-inflammatory properties from *Arthrospira/Spirulina*, *Chlorella vulgaris*, *Chlorella pyrenoidosa*, I*sochrysis*, *Pleurochrysis carterae*, *Dunaliella*, *Porphyridium purpureum*, and *Rhodosorus marinus* [29,35,36,37,38,39]. 

Immunomodulatory agents refer to compounds that increase (immunostimulators) or decrease (immunosuppressants) the response of the immune system. In cosmetology, topical immunomodulators are used to regulate the local immune response of the skin to treat a wide range of skin diseases [40,41]. Moreover, topical immunomodulators are easier to apply and safer for longer periods in comparison with systemic immunomodulators [40,41,42]. 

Anti-inflammatory, antitumoral, antimicrobial, and antiviral properties from a wide variety of compounds are directly linked to immunomodulator activities as they work synergistically with the immune system to elicit an anti-inflammatory response against damaged tissue or act against external microorganisms, interfering with their growth and activating apoptotic cell death. All of these properties have made microalgae metabolites feasible for use as immunotherapy to treat some of these disorders, as mentioned before [28,29,30].

The immune system coordinates its response via multiple signaling pathways, and these responses when unregulated contribute to the pathogenesis of chronic inflammatory skin diseases [43]. The immunomodulator mechanisms include cytokines, interferons, interleukins, and tumor necrosis factors, secreted primarily by macrophages, lymphocytes, and keratinocytes in the epidermis [28,44]. Additionally, various factors such as uncontrolled responses to pathogens, toxic substances and reactive oxygen species (*ROS*)-mediated oxidative stress promote the release of pro-inflammatory mediators [28,45]. Accordingly, immunomodulatory agents are used to inhibit inflammatory responses and fight diseases. 

Immunomodulatory compounds such as exopolysaccharides can interact with cell surface receptors such as the Dectin-1 receptor, Toll-like receptors (*TLR*s), and scavenger receptors in immune cells such as macrophages, dendritic cells, neutrophils, and NK cells [4,46]. Exopolysaccharides and sulfated polysaccharides can bind to these cell surface receptors and induce signaling cascades that activate macrophage, NK cell and T/B lymphocyte activity, phagocytosis, and cytokine secretion [46,47,48,49]. 

*In vitro* and *in vivo* studies have shown the immunomodulatory and anti-inflammatory potential of microalgae-derived bioactive compounds and extracts [50,51,52]. In vitro studies of sulfated polysaccharides from red microalgae *Porphyridium* showed that the polysaccharide inhibited the movement of polymorphonuclear leukocytes, producing an anti-inflammatory effect, while in vivo studies showed that the topical application of the polysaccharides in humans inhibited the development of erythema [51]. Furthermore, immunostimulatory activity has been observed from in vitro assays involving microalgal material such as *Spirulina platensis* extracts, revealing an increase in human dermal fibroblast cell proliferation and enhanced wound area closure rates [52]. Table 1 lists some of the immunomodulatory compounds obtained from microalga and their mechanisms of action, reported extraction methods and culture conditions. 

Studies are currently being conducted with the aim of identifying microalgal compounds responsible for immunomodulatory activities, including mainly the antimicrobial, antidiabetic, anticancer, and anti-inflammatory activities. Some of these studies are described briefly below.

The antibacterial activity of microalgae active compounds has been tested largely for the widespread microbial resistance to antibiotics in a search for new treatments against pathogens. Guzmán et al. [53] identified compounds produced by *Tetraselmis suecica* with antimicrobial properties, finding that an elution of 40% acetonitrile had the highest protein concentration and antibacterial activity tested against three Gram-negative and four Gram-positive bacterial strains. Likewise, in 2020 Alsenani et al. [54] screened 14 microalgae and two cyanobacterial strains to determine their antimicrobial capabilities. The fatty acid methyl esters (FAME) species found in the extracts of *I. galbana*, *Scenedesmus* sp. NT8c and *Chlorella* sp. FN1 showed high inhibitory activity against the growth of all six Gram-positive bacteria tested. Similarly, a study by Mukherjee et al. [55] with the hexane and chloroform extracts from *Scenedesmus obliquus*, tested in both Gram-positive and Gram-negative bacteria, showed the highest antibacterial activity with a minimum inhibitory concentration (MIC) of 15.6–125 μg·mL^−1^. In the same line with previous works, Cepas et al. [56] utilized lipids from over 600 microalgae and cyanobacteria species to determine their activities as antimicrobials and antibiofilms. The authors found that the extracts were effective against one of the three bacterial strains tested, whereas the assayed extracts from the methanol and ethyl acetate fraction reached an 80% biofilm inhibition. In addition, Potocki et al. [57] analyzed the water and ethanol extracts from *Planktochlorella nurekis* against three strains of Gram-positive and two strains of Gram-negative bacteria. They concluded that lauric acid, myristic acid, and stearic acid from the extracts had a high impact on the growth of Gram-negative bacteria. In addition, monounsaturated (*MUFA*) and polyunsaturated fatty acids (*PUFA*) are responsible for the modulation of Gram-positive bacteria.

Another immunomodulatory property of microalgae compounds is their anticancer activity. For instance, Marrez et al. [58] determined the antimicrobial and cytotoxicity activities of crude and fractions from the extract of *Scenedesmus obliquus*. The authors concluded that the microalgae diethyl ether extract had the major activity against bacteria, fungi, and three cancer cell lines, conferring to the microalgae with great potential for use in this field. Correspondingly, Peraman and Nachimuthu [59] evaluated 10 microalgal species for the production of fucoxanthin, and the extracts were tested to determine their antibacterial, antifungal, and antioxidant activities. The microalgae *Dunaliella salina* showed the best activity against Gram-negative bacteria and fungi, whereas for the antioxidant activity, eight of the 10 microalgal strains showed more than 50% inhibition by 2,2′-azino-bis (3-ethylbenzothiazoline-6-sulfonic acid) ABTS scavenging activity. Lastly, Lauritano et al. [60] screened 32 microalgal strains and analyzed the extracts to determine their antioxidant, anti-inflammatory, anticancer, anti-diabetes, antibacterial, and anti-biofilm activities. The authors assessed that from the microalgal species tested, three showed anti-inflammatory activity, one species had anticancer activity, two presented antibacterial activity, and the other two hindered the formation of biofilm.

## 3. Antioxidants

Antioxidants can be defined as “natural or synthetic substances that may prevent or delay oxidative cell damage caused by physiological oxidants having distinctly positive reduction potentials, covering *ROS*/reactive nitrogen species (*RNS*) and free radicals” [61]. This is one of the most common causes of oxidative damage due to *UV* exposure [62]. Lipids, nucleic acids, and proteins are *ROS*, RNS, and reactive sulfur species (*RSS*) targets [63]. *UV*-A damages *DNA* indirectly by generating *ROS*-like radical singlet oxygen (1O2) and/or hydrogen peroxide among others, causing *DNA* mutations during the replication process such as the guanine–thiamine (G–T) transversion, while UV-B directly damages *DNA* strands, generating cyclobutane pyrimidine dimers (*CPDs*) by covalently bonding two neighboring pyrimidines; their accumulation over time can produce an interruption of *DNA* replication and transcription, disturbing the function of the damaged cell [62,64]. Over the last two decades, studies have been conducted to find and obtain antioxidants from natural sources in order to replace synthetic sources to treat oxidative damage. Thus, various studies have focused on the protective effect of enzymatic extracts from microalgae against *DNA* damage induced by oxidative stress [65,66,67,68].

Antioxidants produced by microalgae are substances with high nutritional value, considering that these microorganisms have a greater ability to produce them compared to those obtained from plant-derived sources [69]. They can produce multiple components for a single species, and even if their production yield is lower than those obtained synthetically, nowadays, the generation of bioactive molecules from microalgae is expected to surpass synthetic sources, given that their production is renewable and sustainable [70]. Some of the antioxidants that can be obtained from microalgae are chlorophyll, vitamins, flavonoids, polyphenols, sterols, and carotenoids [15,63]. Table 2 lists a variety of microalgae-derived products with antioxidant properties. Carotenoids, β-carotene, chlorophyll a, chlorophyll b, and xanthophylls were the main photosynthetic pigments identified in green microalgae such as *Spirogyra neglecta* and *Microspora indica* [71]. Fucoxanthin, a bioactive microalgal compound, can be found in several species of diatoms (golden-brown microalgae) and has remarkable antioxidant, anti-obesity, anti-diabetic, anti-cancer, anti-inflammatory, anti-hypertensive, and anti-osteoporotic properties. Fucoxanthin is an important carotenoid for human health that is marketed at USD 30,000 per gram and is considered a valued nutraceutical product or functional food to prevent or help treat different diseases [72,73,74]. Recent research on the use of diatoms rich in carotenoids has been extended to space missions where resources are limited and the need to implement CO_2_ recycling systems provides an opportunity to produce O_2_ and food based on microalgal biomass as an alternative to health supplementation for humans in these conditions [69,75,76]. Additionally, in vitro experiments on human fibroblasts have reported that even small doses of infrared radiation can produce free radicals, affect collagen and elastin expression, and upregulate metalloproteinases (*MMP*s), representing an area of opportunity for the development of alternative supplementation to obtain antioxidants [77].

Microalgae can produce a large variety of vitamins, including, vitamins A, B1, B2, B6, B12, C and E. Minerals can also be obtained from microalgae, such as potassium, iron, magnesium, calcium, and iodine, in addition to the high protein content with complete essential amino acids [69,78]. *Dunaliella tertiolecta* synthesizes vitamin B12, B2, E, and beta carotene [79]. *Tetraselmis suecica* produces vitamin C, a *ROS* scavenger, and protects against lipid peroxidation via lipid hydroperoxyl radical reduction [80]. extract is rich in carotenoids (xanthophylls, lutein, violaxanthin, neoxanthin, *T. suecica* antheraxanthin and loroaxanthin esters), and it has shown excellent antioxidant, antiproliferative, and repairing activity in human lung cancer cell lines [81]. Pistelli et al. previously reported that *Skeletonema marinoi*, *Cyclotella cryptica*, and *Nannochloropsis oceanica* coculture improved bioactive compound richness as vitamins and biomass augmented the antioxidant or chemoprotective activity [82]. The *Skeletonema marinoi* strain has also been studied for ovothiol biosynthesis, an efficient thiohistidine compound in the scavenging of radicals and peroxides and which has only been identified in clams [83]. Astaxanthin (a red secondary carotenoid) is denominated as a super antioxidant, and most of its production is carried out synthetically (95%); while the microalgae *Haematococcus pluvialis* is the natural source for this excellent compound. It is present in other organisms such as plants, yeast, bacteria, seafood and other microalgae, but it is in *H. pluvialis* that the 3S,3′S stereoisomer is produced, and it is the most profitable of the three ((3S, 3′S); (3R, 3′S), and (3R, 3′R)) and 65 times more powerful than vitamin C and 54 times stronger than β-carotene in antioxidant activity, highlighting that the synthetic form is 20 times less effective in antioxidant quality than the natural form [84]. Glutathione scavenges electrophilic and oxidant species either in a direct way or through enzymatic catalysis: (i) by directly quenching reactive hydroxyl free radicals, other oxygen-centered free radicals, and radical centers on *DNA*, and (ii) as the co-substrate of glutathione peroxidase, allowing peroxide reduction [85]. However, glutathione also plays an oxidant role to a lesser extent, during GSH catabolism [85]. Mycosporine-like amino acids (*MAA*s) have the ability to scavenge *ROS* and can be of significant in the scavenging of free radicals induced by UV radiation [86].

Antioxidants are synthesized by microalgae commonly as a response to environmental stress, mainly to avoid oxidative stress [63]. The microalgae-derived antioxidants can be used in cosmeceutical applications such as moisturizers and sunscreens to prevent and treat multiple skin conditions from photoaging to skin cancer [87,88,89].

**Table 2 molecules-27-03512-t002:** List of compounds with antioxidant activity obtained from microalgae.

Type of Compound	Mechanism of Action	Microalgae Species	Culture Conditions	Extraction	Reference
Amino acids	*ROS* attenuation	*Chlamydomonas hedleyi* (Chlorophyta)	F/2 medium, 22 ± 1 °C, 80 μmol·m^−2^·s^−1^ light, 16:8 D:L cycle rotary shaker + bubbled with CO_2_-enriched (1%) air.	Digestion of dry biomass with aqueous methanol (20% *v*/*v*) at 45 °C.	[90]
Polysaccharides	Attenuation of free radicals, hydroxyl radicals, and *ROS*	*Arthrospira platensis* (Cyanobacteria)	Zarrouk medium, 32 ± 1 °C, 100 μmol·m^−2^·s^−1^ light, stirred with CO_2_-enriched (1%) air.	Tangential ultrafiltration cell—30 Kda membrane	[91]
	Free radical scavenging	*Odontella aurita* (Bacillariophyta)	Modified L1 medium (6–18 mM), 25 ± 2 °C, 100 and 300 μmol·m^−2^·s^−1^ light, sparging with air + 1% CO_2_.	Digestion of dry biomass at 60 °C withsulfuric acid	[92]
Sulfated polysaccharides	*ROS* attenuation	*Porphyridium* sp. *(*Rhodophyta*)*	Seawater medium, 24 ± 3 °C, 150 μmol·m^−2^·s^−1^ light, bubbled with air + 1–3% CO_2_.	Culture centrifugation (17,000× *g*, 20 min) and supernatant filtrate in dialysis tube (⌀ 2.3 cm—MW cutoff 8000)	[93]
Extracellular polysaccharides	Free radical scavenging	*Rhodella reticulata* (Rhodophyta)	Kock medium, 25 °C, 92 μmol·m^−2^·s^−1^ light.	Ethanol extraction	[94]
Pigments, peptides and vitamins	*ROS* attenuation	*Skeletonema marinoi* (Ochrophyta)	F/2 enriched medium, 20 °C, 150 μmol·m^−2^·s^−1^ light, 12:12 D:L cycle.	Mechanical grounding with absolute methanol	[95]
Pigments	Free radical scavenging	*Dunaliella salina* (Chlorophyta)	Johnson medium with artificial seawater (35 g L^−1^), pH 7.5, 100 μmol·m^−2^·s^−1^ light, 12:12 D:L cycle, bubbled with air 2 L min^−1^.	Sonication with methanol and filtration Fluoropore PTFE 0.2-μm	[96]
	*ROS* attenuation	*Chlorella vulgaris* (Chlorophyta)	BG-11 medium (without and with nitrogen starvation + *NaCl* addition 30%), 25 °C, 150 and 1000 μmol·m^−2^·s^−1^ light, bubbled with air.	Homogenization with acetone and supernatant filtration with Na_2_SO_4_	[97]
	Free radical scavenging	*Anabaena vaginicola* (Cyanobacteria)	BG-11 medium without nitrates, 400 μmol·m^−2^·s^−1^ light, 25 ± 2 °C, 12:12 D:L cycle	Freezing of biomass dry at −20 °C with methanol + ultrasonication	[98]

No extraction methodology reported. D:L (dark:light) cycle.

## 4. Photoprotectors

As mentioned before, microalgae possess mechanisms against environmental challenges such as activation of several photo/dark repair mechanisms, antioxidant systems, *UVR* avoidance, *DNA* repair, and cell protection by producing *UV* photoprotective compounds such as mycosporine-like amino acids (*MAA*s), carotenoids, polyamines, and scytonemin [99].

Ectoine is a compatible solute that serves as a protective substance by acting as an osmolyte and thus helps organisms survive extreme osmotic stress, for example, with high salt content [100]. Ectoine from *Halomonas* elongate extracted with low salt concentration solution is an adsorbent of ultraviolet light [101,102]. Mycosporine-like amino acids that also absorb ultraviolet light can be extracted from dry biomass employing methanol to assist the extraction and chloroform to eliminate pigments from *Scytonema cf. crispum* [103].

The variation in different growth parameters has been associated with *MAA* composition in microalgae, due to improved production in direct correlation with sunlight exposure, or a variation in the *MAA*s’ accumulation according to the type of light [104]. As there are diverse molecular configurations and growth conditions of the harvested microorganisms, the spectrum of light that the photoprotectors can adsorb is also diverse. Photoprotection is defined as the decrease in *UV* radiation damage that can cause skin disease and the risk of skin cancer [105]. *UV* radiation comprises *UV*A (320–400 nm), *UV*B (290–320 nm), and *UV*C (200–290 nm). *UV*A causes indirect *DNA* damage via *ROS* production; it is also related to skin aging and pigmentation. *UV*B exposure enhances sunburns and *DNA* strand breaking; the pyrimidine dimer mutations associated with nonmelanoma skin cancer are also linked to *UV*B exposure. *UV*C, the *UV* radiation with higher energy, is completely absorbed by the ozone layer and does not represent a risk from its natural source [106].

Scientific research and new product development efforts in photoprotectors have been pursued. Some *UV*-resistant compounds have been investigated for their potential application in the generation of new products. Lutein, a compound that protects the skin from damage caused by *UV* rays, has been found in different microalgae such as C. protothecoides, Scenedesmus almeriensis, *Muriellopsis* sp., *Neospongiococcus gelatinosum*, *Chlorococcum citriforme, C. zofingiensis*, *D. salina*, and *Galdieria sulphuraria* [27]. The application of *MAA*s in the cosmetic field has also been studied previously since these compounds can absorb light between 309 and 362 nm and dissipate radiation as heat, protecting cells from mutation caused by *UV*-R and free radicals [107]. Some microalgae that have been reported to contain this compound include *Anabaena* spp., *C. vulgaris*, *D. salina*, *Eutreptiella* sp., *Scenedesmus* sp. and S. platensis. High accumulations of carotenoids in microalgae can also provide photosynthetic protection features [108]; this effect has been reported in the microalgae *Nostoc* sp., *Eutreptiella* sp., *C. protothecoides*, *P. antarctica* and *P. glacialis*.

There are not many reports on the application of the photoprotective properties of microalgae in cosmetic products such as sunscreens. Nevertheless, this is a finding that can generate substantial impact in the cosmetics market for *UV* skincare products. It has been reported that for some microalgae, *MAA*s also play a role as anti-desiccants that help the cell to overcome nocive effects from *UV*-B radiation; a study reported this behavior for the *Leptolyngbya* sp. cyanobacteria [109].

The study of microalgae and cyanobacteria responses against *UV* radiation in a combination of other conditions is an approach covered by many authors, some of them are described as follows. Singh et al. in 2020 evaluated the responses of *Anabaena* sp. in comparison to *UV*B radiation and exogenous ammonium chloride as a form of nitrogen supplementation. The study probed an interaction between *NH_4_Cl* supplementation with a protective effect, according to the photosynthetic activity, maximum quantum efficiency of *PS*II, and maximum electron transport rate. The MAAs also were accumulated in larger quantities in relation to *NH_4_Cl* supplementation [110]. On the other hand, *C. vulgaris* was evaluated for the capacity to resist different *UV*-B radiation intensities and times of exposure, regarding inhibition of growth by 50% at the maximum intensity and time evaluated. The study showed that the exposure of *C. vulgaris* to short-time periods or low-intensity levels enhanced the production of carotenoids [111]. Additionally, a comparative study of *C. vulgaris*, Microcoleus vaginatus, *Nostoc* and *Scytonema javanicum* distinguished the most competent microalgae with regard to *UV*B radiation due to *ROS* attenuation and their capability to repair photosystem II and *DNA* damages. Among the different species, *Nostoc* sp. was the strain capable of surviving the high levels of radiation [112].

Active compounds such as scytonemin along with *MAA*s are determinants that protect cyanobacteria from *UV* damage [113]. The biosynthesis mechanism of scytonemin has been recently elucidated in a model strain Nostoc punctiforme due to the promising antioxidant and *UV* protection properties that represent a potential for cosmetic and medical applications [114]. In a similar approach to Singh et al. 2020, Bennett and Soule 2022 evaluated osmotic stress and *UV*A and *UV*B exposure in the *N. punctiforme* strain to evaluate the expression of scytonemin genes. The study reported an increase in scytonemin gene expression in the presence of *UV*A, *UV*B and high light; however, the up-regulation of the genes did not reflect the scytonemin production [115]. Another study in 2021 was carried out to evaluate the physiological changes and scytonemin production due to UV radiation and salinity in cyanobacteria; as *ROS* increased, the protein and phycobiliprotein contents were reduced. The exposure to photosynthetically active radiation (*UV*A and *UV*B) and osmotic stress for at least 3 days induced scytonemin production in *Scytonema* sp. strain [116]. Orellana et al. in 2020 reported increases in scytonemin production in the desertic area of Atacama, showing that the indigenous endolithic cyanobacteria (*Halothece*) from Salar Grande suffered scytonemin reduction due to *UV*-A radiation under the simulated desertic conditions [117]. Table 3 shows a list of microalgae that have shown *UV* resistance or have been tested to exhibit *UV* photoprotective compounds.

## 5. Moisturizers, Regenerative and Other Activities

In the dermis, collagen has a dual-action mechanism providing building blocks for the formation of collagen and elastin fibers, while also binding to receptors present on fibroblasts to stimulate the production of collagen, elastin, and hyaluronan, which helps to maintain good skin appearance and elasticity and enhance its strengthening against harmful environmental factors [124]. The bioactive compounds obtained from microalgae sources have proven to possess properties like the ones presented in this paper, providing a range of applications that vary widely depending on the industrial and commercial sector of interest [125]. Expressly, bioactive compounds from microalgae are currently being incorporated into the cosmetic and cosmeceutical industry as they can improve and maintain the structure and function of the skin [1].

Exopolysaccharides from microalgae can be rheology modifiers, conditioners, moisturizing agents, healing agents, emulsifiers, and substitutes for hyaluronic acids. In addition, they can stimulate collagen synthesis and provide protective activities against enzymatic proteolysis [31,32,33,124]. Proteins found in microalgae used as food supplements can stimulate collagen synthesis, leading to a reduction in vascular imperfections and promoting tissue regeneration [1,33,126,127,128]. Additionally, some microalgae species are known to produce compounds that can act as substitutes for hyaluronic acids or protect against enzymatic proteolysis [1,31,32,33,126,127,128,129]. Table 4 shows some of the main molecules obtained from microalgae cultures that could be implemented as moisturizers and promote putative regenerative activity on the skin.

Furthermore, chlorophyll a and b can be used as dyeing agents and also as additive agents that can be used to mask odors in formulations [39,130]. β-carotene, phycoerythrin, phycocyanin, allophycocyanin, phycoerythrocyanin, astaxanthin, lutein, lycopene, and violaxanthin can be used as dyeing agents in the textile industry and in cosmetics as greener alternatives [1,130,131].

Maintaining correct skin moisturization is the first step to aiming for a strong defense mechanism against irritant agents and tensioactive materials. It has been clear that during the actual COVID-19 pandemic emergency, several alcohol-based products released as sanitizers also help to reduce the transmission of the virus. Nonetheless, those products often need to be supplemented with humectants, emollients, or moisturizers to avoid the effects of alcohol, which principally affect the epidermal layer, causing dehydration. Although dry skin does not represent a serious problem, it can lead to complications in the management of dermatological infections [132,133].

**Table 4 molecules-27-03512-t004:** List of compounds with moisturizing and regenerative activity obtained from microalgae.

Type of Compound	Mechanism of Action	Microalgae Species	Culture Conditions	Extraction	Reference
Collagen	Stimulation of collagen synthesis in the skin	*C. vulgaris*	NR	Methanol extraction	[134]
	Regeneration by stimulation of collagen synthesis in the skin	*S. platensis*	Zarrouk’s medium, pH 9.8–10.0, 25 °C, 46 μmol m^−2^·s^−1^ light	Raw biomass	[135]
Mycosporine-2-glycine	Collagenase inhibition and glycation products inhibition	*Aphanothece halophytica*	BG-11	Methanol extraction and mechanical disruption	[136]
Polysaccharides	Moisturizing agents	*Pediastrum duplex*	MIII medium, pH 7.9 ± 0.1 20 ± 0.5°C, 80 µmol m^−2^·s^−1^ light, 12:12 D:L	Methanol extraction	[137]
	Regulate water distribution in the skin	*Codium tomentosum*	NR	NR	[138]
	Moisturizing agents	*Undaria pinnatifa*	NR	NR	[138]
	Moisturizing agents	*Durvillea antarctica*	NR	NR	[138]
	Moisturizing agents	*Cladosiphon okamuranus*	NR	NR	[138]
Fat acids	Dermal collagen content rescue	*S. rubescens*	NR	Hydrophilic hot extrusion	[139]
Amino acids	Increases expression of Procollagen C Proteinase	*C. hedleyi*	F/2, pH 8–9, 22 °C, 80 μmol m^−2^·s^−1^ light, 8:16 D:L cycle, 1% CO_2_	Raw biomass	[91]
TrpA protein	Collagen-like protein	*Trichodesmium erythraeum*	NR	NR	[11]

D:L = (dark:light) cycle. NR = Not Reported.

## 6. Novel Vehicles or Excipient Compounds

Some microalgae-derived molecules can serve as vehicles or excipients for other molecules of interest. Proteins from microalgae are molecules of commercial interest; they can be used in the food industry, pharmaceutical, and cosmetic sectors, or more specifically, in nutraceuticals and cosmeceuticals. They can be used as emulsifying, foaming, thickening, and gelling agents [15]; these functions are mainly explored in the food sector but could potentially be extrapolated to other sectors. *C. vulgaris* and *Tetraselmis* sp. have emulsifying and foaming properties. Thickening or gelling properties have also been reported for *Arthrospira platensis*, *H. pluvialis* and *T. suecica proteins* [140]. However, this application field is recent; therefore, not many studies have been conducted in this area.

Extracellular polysaccharides (*EPS*s) are secreted by various marine organisms, including microalgae. They represent a physical barrier protecting cells from harmful agents and environmental constraints and serve in different physiological processes including cell adhesion, cell interaction, and biofilm formation. The protective capacity of microalgae *EPS*s has been reported also in the environment, where diatoms delay the degradation of materials through the secretion of *EPS*s that allows adhesion to the material superfice and create a barrier between the material and the corrosive environment. This behavior can be extrapolated to the potential use of these biomolecules in mixtures with topical bioactive compounds for slow release into the skin or for the improvement of the skin protection capacity in commercial products. Other areas of interest for the use of *EPS*s are the pharmaceutical and medical industries, where properties such as viscosity, porosity, and absorption capacity of *EPS*s produced by microalgae and cyanobacteria can be applied for the development of different products such as drug delivery hydrogels, tissue engineering, occlusion devices, and bioadhesives for active release, among others [141]. One of the main strategies to improve *EPS* applications is the management of *EPS* rheological properties, which have been reported to depend on EPS concentration. In this sense, the characteristics can be controlled in order to allow their use in different applications as active texturizing agents, carriers, or as active protectors and enhancers [142]. Furthermore, *EPS*s have been reported as important emulsion stabilizers, flocculants, foam stabilizers, and hydrating agents (tested in cosmetics and pharmaceuticals) [143]. According to the literature reviewed, the applications of microalgae *EPS*s as vehicles or excipient compounds showed interesting characteristics to impact a wide range of options. Table 4 is lists compounds obtained from microalgae with applications in topical formulations.

Extracellular vesicles (*EV*s) are membrane-surrounded structures released in the extracellular milieu by every cell type. *EV*s contain proteins, lipids, nucleic acids, and other metabolites [144]. In addition, EVs possess many attributes of a drug delivery vehicle of interest to the cosmeceutical field. A newly discovered subtype of *EV*s derived from microalgae, named nanoalgosomes, are extracellular nano-objects from cultures of microalgal strains. Nanoalgosomes are novel membranous biogenic nanomaterials refined for the first time from a sustainable and renewable bioresource (microalgae), which can be used as a new natural delivery system for high-value microalgal substances (such as antioxidants, pigments, lipids, and complex carbohydrates), bioactive biological molecules and/or synthetic drugs. *EV*-like nanoparticles have been isolated from several microalgae strains such as *T. chuii* and *D. tertiolecta* [145], suggesting that other microalgae may contain *EV*-like nanoparticles and have a major impact on drug delivery in the cosmeceutical field.

Research has been conducted regarding the potential of porous silica-based particles for drug delivery applications due to their extended drug release profiles and high efficacy in delivering hydrophobic drugs [146]. These silica-based nanoparticles obtained from diatoms are used as drug delivery carriers due to their biodegradability, easy functionalization, low cost to obtain and maintain, and simple features compared to the synthetic ones, which make these agents proper alternatives to synthetic silica nanoparticles [147].

These findings demonstrate the potential of microalgae as excipients and vehicles for molecules or drugs, implying great potential for application and a breakthrough in the search for more natural alternatives in cosmeceuticals.

## 7. Perspectives

Currently, microalgae cultures as a source of high-value biomolecules have become more attractive due to their potential application in the development of technologies to mitigate greenhouse gas emissions, which agrees with the sustainable development goals of the United Nations. Nevertheless, most of the bibliography consulted microalgae biomass is cultivated using synthetic and controlled media. In this sense, future research should be focused on the evaluation and optimization of processes that allow the fixation of CO_2_ from different sources at the same time that microalgae produce molecules of interest. Actually, the obtention of molecules from microalgae biomass have been widely studied, and reported data showed a dependence of the produced metabolite profiles on the microalgae strains and different stress conditions (light intensity, temperature, pH, nutrients concentrations, and others). For future research, the authors identified a clear procedure for the obtention of biomolecules from microalgae that can be described in four main steps: (1) microalgae strain selection, (2) standardization of growth conditions, (3) biomolecule characterization, and (4) extraction and functionalization (Figure 2). The photobioreactor design, moreover, represents an area of opportunity to improve those processes, to guarantee the light incidence to the microalgae culture, the retention time for gasses, and small spaces for technology implementation in established industries. Finally, the extraction and purification process represents a challenge due to the different nature of those metabolites of interest and the characteristics of the different microalgae. Developing simpler extraction protocols is also an area for improvement. However, efforts should be oriented to the implementation of processes where microalgal compounds are more accessible and can be used directly from raw biomass or biomass that requires minimal processing.

Along with microalgae, macroalgae (not explored in this work) also represent an alternative source of molecules of interest to improve functionality and extend the time of action. Furthermore, biocompatibility analysis of those metabolites must be characterized to ensure their safe use as a cosmeceutical components.

Unfortunately, most of the biotechnology research only focuses on biomolecule extraction with certain putative activity from microalgae cultures, as shown by some of the data presented in this work. Nevertheless, microalgae can produce a wide variety of biomolecules that can be approached in different areas, so there is still work to be done to bring biodegradable microalgae-based products into general use.

The investigation of the presence of cosmeceutical compounds in microalgae has been widely reported; however, despite the existence of records on the extraction and evaluation of the potential of these compounds, the information regarding their application and distribution in the market remains scarce, which could be an indicator of the lack of scaling of this practice from a laboratory test to its commercial application. It is necessary to evaluate strategies that allow scaling the use of these biomolecules, and to identify barriers to this process. Is it the quality of the obtained biomolecules? Is it a scarce market for the product? Does misinformation about the advantages and disadvantages exist? Or, are there limitations on access to economic resources? Regardless of the answer, the next steps for the use of microalgae bioactive compounds should consider a market study that allows the materialization of these areas of opportunity.

## 8. Conclusions

The cosmeceutical industry has been expanding over the years and making an effort to use active ingredients from natural sources. Microalgae are a source of added-value functional compounds that can be ultimately a healthier option. Moreover, microalgae-derived metabolites can be obtained through low-cost, eco-friendly cultivation processes. The application of microalgae as a source of bioactive molecules has been successfully evaluated by different authors. In the process of taking advantage of the potential of microalgae-derived compounds to develop and improve a new generation of cosmeceutical products, it is important to consider the microalgae strain, growth conditions, biomass characterization, and protocols for extraction and functionalization. Those cosmeceutical products enriched with photoprotectors, antioxidants, immunomodulators, and moisturizers/regenerative molecules extracted from different microalgae species suggest an improvement in the potential of different cosmeceutical product characteristics with novel advantages in terms of environmental impact.

Most of the actual studies develop microalgae growth under specific stress conditions to stimulate the production of the metabolites of interest for posterior extraction and use in the development and enrichment of cosmeceutical products. Nevertheless, the direct application of raw or dry microalgae in the process of product improvement is still scarce; hence, the use of microalgae as a source of bioactive molecules still has areas of opportunity for future research, focused on the method of approach and its application to commercial cosmeceutical products.

## Figures and Tables

**Figure 1 molecules-27-03512-f001:**
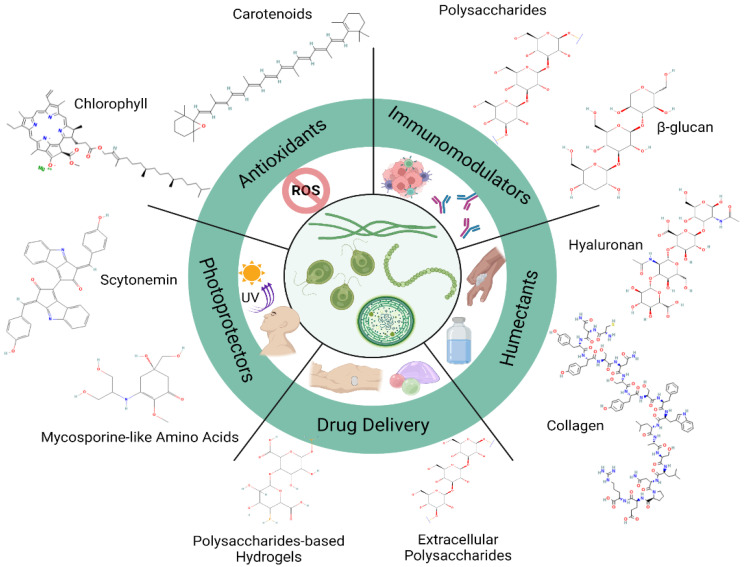
Molecular structure from microalgae-derived compounds used for multiple biotechnological applications.

**Figure 2 molecules-27-03512-f002:**
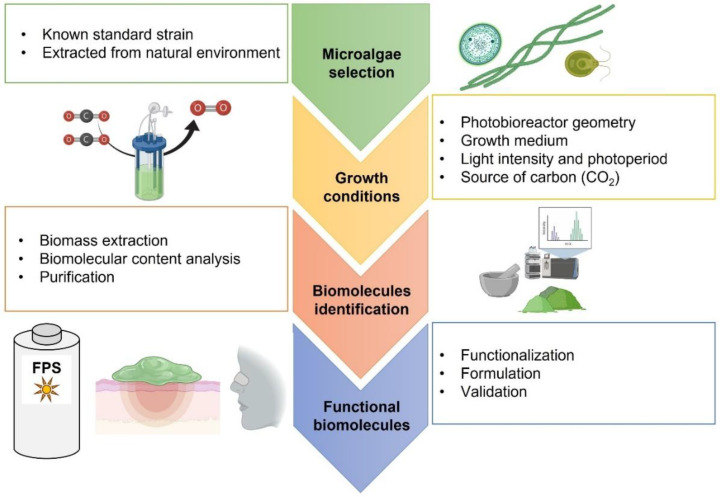
Microalgae-derived biomolecule production. Four main steps are required to obtain biomolecules from microalgae biomass: (1) microalgae selection, (2) culture standardization, (3) purification, and (4) functionalization. Considerations must be made for maintaining sustainability and efficacy. Created with BioRender.com.

**Table 1 molecules-27-03512-t001:** List of compounds with immunomodulatory activity obtained from microalgae.

Type of Compound	Mechanism of Action	Microalgae Species	Culture Conditions	Extraction	Reference
Peptides	Antimicrobial activity through interaction with negatively charged membranes	*T. suecica* (Chlorophyta)	F/2 medium, 21 ± 0.5 °C and 40 W Light	Acid extraction and reverse phase column chromatography separation.	[53]
Fatty acid methyl esters (*FAME*), lipids and carotenoids	Interference in biosynthesis of bacterial fatty acids	Various microalgae species	F/2 medium for marine microalgae and Bold’s basal medium (*BBM*) for freshwater microalgae	Three extraction systems with different solvents	[54]
Fatty acids and pigments	Interference in biosynthesis of bacterial fatty acids	*S. obliquus*	-	Polarity-wise successive solvent extractions	[55]
Lipids	Membrane permeabilization	*S. brasiliensis*, *E. acutiformis* and *Sphaerospermopsis* sp.	M7 medium for microalgae, Z8 medium for cyanobacteria	Extraction with different solvents	[56]
Fatty acids	Interference in biosynthesis of bacterial fatty acids	*P. nurekis*	Medium with various macro- and microelements at pH < 6, 30 °C and 16,000 LUX	Extraction with water and ethanol	[57]
Fatty acids and phenolic compounds	Interference in biosynthesis of bacterial fatty acids	*S. obliquus*	BG-11 medium	Extraction with organic solvents and sonication	[58]
Pigments	Free radical scavenging	Various microalgae species	Guillard’s F/2 medium, 22 ± 2 °C, 14:10 D:L cycle, 60–65 µ·E·m^−2^·s^−1^light for 16 days	Ethanol extraction	[59]
Microalgae extract	Various mechanisms tested	32 microalgae species	Guillard’s F/2 medium, 19 °C, 12:12 D:L cycle, and 100 µmo·m^−2^·s^−1^ Light	Sonication and acetone extraction	[60]

**Table 3 molecules-27-03512-t003:** List of different microalgae with photoprotection features.

Microalgae Species	UV Test	Intensity Unit	Resistance Factor	Type of Study	Culture Conditions	Reference
*Anabaena* spp. (Cyanobacteria)	*UV*B	1 W·m^−2^ for 4 h/day	Scytonemin, MAAs	Exposure to *UV*B-R	BG-11 medium, *NH_4_Cl* as *N*-source, 28 ± 2 °C	[110]
*Characium terrestre* (Chlorophyta)	*UV*B	NR	Sporopollenin	Exposure to extreme *UV*B irradiance	Medium reported by Zachleder et al., 26 °C.	[118]
*C. protothecoides* (Chlorophyta)	*UV*C	0.01 to 0.20 W·m^−2^	Lutein	Exposure to different *UV*C irradiances	BG-11 medium, 25 ± 1 °C	[119]
*C. vulgaris* (Chlorophyta)	*UV*B	1 to 5 W·m^−2^	Sporopollenin, Scytonemin, *MAA*s	Exposure to different *UV*B irradiances	Bold Basal medium, 25 ± 1 °C	[111]
*D.salina* (Chlorophyta)	*UV*A *UV*B	110 mmol·m^−2^·s^−1^ for *UV*A	Sporopollenin, Scytonemin, *MAA*s	Exposure to *UV*A-R and *UV*B-R	Medium with NaCl, 26 °C	[108]
*Eutreptiella* sp. (Euglenozoa)	*UV*B	280 to 320 nm	Xanthophylls, *MAA*s	Tested under fixed light	F/2 medium, 10 °C	[120]
*Nostoc* sp. (Cyanobacteria)	*UV*B	312 nm	Carotenoids, Scytonemin	Photosynthetic activity essay	BG-11 medium, 25 °C	[112]
*O. aurita* (Bacillariophyta)	*UV*A *UV*B	110 kJ·m^−2^	D1 protein, activation of antioxidant enzymes	Exposure to *UV*A-R and *UV*B-R	Artificial seawater reported by Harrison et al.	[121]
*N. sphaeroides* (Cyanobacteria)	*UV*A	320 nm	Not identified	Exposure to *UV*A irradiance	BG-11 medium, 23 °C	[122]
*P. antarctica*, *P. glacialis* (Rhodophyta)	*UV*B	280 to 400 nm	*MAA*s, Xanthophylls	Acclimation to photosynthetically active radiation	GP5 medium, 1.0 ± 0.5 °C	[123]
*S. platensis* (Cyanobacteria)	*UV*A	320 nm	Sporopollenin, Scytonemin, *MAA*s	Exposure to *UV*A irradiance	BG-11 medium, 23 °C	[122]
*N. punctiforme*	*UV*A *UV*C	100 µm photon m^−2^s^−1^	Scytonemin	Exposure to *UV*A, *UV*B and osmotic stress	AA/4 medium	[115]
*Scytonema* sp.	*UV*A *UV*B	6.5 Wm^−2^ 0.56 Wm^−2^	*MAA*s and antioxidant enzymes	Photosynthetically active radiation and salinity	BG-11 medium	[116]
*Halothece*	*UV*A	3.6 Wm^−2^	Scytonemin	Exposure to *UV*A-R	Halites	[117]

## Data Availability

Not applicable.

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
