# Peer review of "Microalgae Bioactive Compounds to Topical Applications Products—A Review"

_molecules, 2022, doi:10.3390/molecules27113512_

Round 1

Reviewer 1 Report

Manuscript Molecules 1603511

This manuscript by Martínez-Ruiz et al is a revision on cosmetics and cosmoceuticals containing microalgae and or natural metabolites isolated from algae. The main classes of bioactive metabolites described in the review are immunomodulators, antioxidants, photoprotectors, moisturizers, and vehicles or excipient compounds. Perspectives and Conclusions are presented. There are 145 references cited, 16 from 2020, 5 from 2021, and 2 from 2022. Since microalgae are well studied organisms, the literature was not well explored, and more papers from the last 2-3 years were expected. On the other side, many interesting papers are cited but only as a reference number in the manuscript. This manuscript does not go deep enough in the proposed theme, and needs a deep revision before being read for submission.

I will point some general points below, but the authors should take the points as examples. The authors did not use the line numbers present in the template, making difficult to write the review.

The manuscript is wordy and the language must be professionally revised.

I was very disappointed to see that, despite the title mentions “bioactive compounds” and “microalgal-derived products”, very few molecules are explored in the manuscript. Chemical structures are not presented.

Some general comments:

The genus should be abbreviated when cited by the second time. (C. vulgaris, for example)

Authors mention secondary metabolites, but the examples contain amino acids, collagen, proteins… Other examples (Table 3) do not refer to the use of metabolites.

I started correcting the language mistakes, but only as examples.

Abstract

Please, rephrase: “Microalgae bioactive compounds identification, extraction, and application” I suggest, for readability: The identification, extraction, and application of microalgae bioactive compounds…

Please, correct: “… literature has focused in on the obtention development of…”

Long sentence: “Most of the literature …. of photoprotectors, antioxidants, phycoimmunomodulators and moisturizers/regenerative molecules extracted from different microalgae species. (add a full stop here and start a new sentence) Identification of these molecules and their application in topical cosmeceutical products showed has shown an increase in the potential of commercial products”.

Please correct: “… and results of implement implementation in cosmeceutical products”

Please, correct: “Aditional In addition to this…”

Page 1: “Cosmeceuticals are mainly lotions or creams …… Their its importance is underestimated”

Page 2:

Please, correct “Other natural sources can be considered because of being they are more eco-friendly, like microalgae.”

Please, replace the comma by a full stop here: “and forms,...”

Please rephrase “These metabolites have the potential to cause a big impact”. They do not have potential, since many of they are already being used.

Please, replace the comma by a full stop after “ … of primary metabolism, several…”

Please, replace the comma by a full stop after “[8],” As you can see, this is a recurring error in the manuscript, long sentences and improper use of punctuation. It is necessary professional language revision.

When you say “They can grow in variate media such as wastewater and leachates as a result of the bioremediation process”, do you mean that we can produce cosmetics/cosmeceuticals growing algae in wastewater? I think bioremediation and biotechnological production of cosmetic ingredients follow different approaches. Ok, algae can grow in wastewater, but it is not safe/helpful for producing cosmetics, which is the theme you are reviewing.

The introduction is very wordy. Please, revise to remove tautology. Some concepts are repeated: “environmentally friendly”, “molecules that can be obtained from microalgae are molecules that can act as cosmeceuticals”

Figure 1. The chemical structures are too small. I suggest adding them outside the circle, so they can be visible.

Item 2. Phycoimmunomodulators. Since in items 3, 4… you do not use Phycoantioxidants, Phycophotoprotectors,…, here you should use Immunomodulators only.

Please, remove “Some of the immunomodulatory compounds that can be obtained from microalgae are exopolysaccharides such as sulfated polysaccharides and β1,3-glucans. [18–20].”, since you repeat almost the same thing in the beginning of item 2.1.

At this point, references 18-20 are cited, which describe important aspects of immunomodulation of metabolites from algae, such as mechanisms of action, current developments, etc. In a review, it is expected that there will be a deepening of the central theme and not only the presentation of a list of references, so the content of the cited literature should be further explored.

Item 2.1 Which is the correlation of anti-inflammatory, antimicrobial and antiviral properties with immunomodulation? Or anti-tumor, anti-apoptotic, etc, as you mention later? This item presents a list of species “produce polysaccharides with antiinflammatory effects”; therefore, it is necessary to explain the relationship between immunomodulation and inflammation.

In item 2.2, you give some information linking immunomodulation and inflammation. I suggest to explain this in item 2 moving items 2.3 and 2.4 to there (before 2.1). Also, it is not necessary to have items 2.3 and 2.4, since there is little information to justify subtitles. Once combining the items, please attention to avoid repetition.   

In addition, you say “In the literature we can find several reports of microalgae compounds with immunomodulatory activity”, but you only show examples of 1,3-β-glucans. The chemical structures of the molecules cited in this revision should be presented.

Item 2.2. “The mechanisms involved in the immunomodulatory effects of some of the afore mentioned compounds ..” Which ones? You cited only a few compounds, please, specify.

Table 1. Please, explain why antiviral and antimicrobial activity are the mechanisms of action of immunomodulatory compounds. Correct the misuse of ccapitals and use the correct symbol for degree. Please, explain what is m−2 s −1

Berthon, J.Y.; Nachat-Kappes, R.; Bey, M.; Cadoret, J.P.; Renimel, I.; Filaire, E. Marine algae as attractive source to skin care. Free Radic. Res. 2017, 51, 555–567, doi:10.1080/10715762.2017.1355550.

  1. Antioxidants

3.1. In reference [57], the authors say “Microalgal biomass is considered to be superior source of nutritional anti-oxidants due to its higher production capacity compared to conventional plant-derived sources.” I think it is not the same thing that the authors wrote.

Also, in reference [58], the authors say “Therefore, microalgae are recognized as an excellent source of natural colorants and nutraceuticals and it is expected they will surpass synthetics as well as other natural sources due to their sustainability of production and renewable nature (Dufossé et al., 2005).” Please, rephrase.

In this part you focus on carotenoids, but there are many other compounds produced by algae with antioxidant potential. Coulombier et al (2021) cite glutathione, flavoides such as phloroglucinol and hydroxycinnamic acid, mycosporins-like aminoacids [Coulombier, N., Jauffrais, T., & Lebouvier, N. (2021). Antioxidant compounds from microalgae: A review. Marine drugs19(10), 549.]. See also Barkia, I., Saari, N., & Manning, S. R. (2019). Microalgae for high-value products towards human health and nutrition. Marine drugs17(5), 304.

Table 2. Correct the same mistakes pointed for Table 1. Correct CO2 formula. BG-11, F/2, etc are culture media? It is difficult to understand, because in some pleces it is clear (for example, Zarrouk medium), but in other places, there is only a code. How “dry biomass” can be an extraction?

Item 3.2. Ok, but, the scope of the review is cosmetics. Which are the specific applications in that field?

  1. Photoprotectors. This item dives just a few examples. It was expected to read about the cosmetic use of the algae species cited in Table 3. Please, revise the table; here the degree symbol is correct, but a space is missing between the value and the degree symbol.

For “artificial seawater”: other conditions should be specified, at least, the temperature.

  1. Moisturizers

In this item, also under-explored, I do not understand why so many papers are cited if their content is not explored.

Table 4. Please, see previous observations.

  1. Perspectives

Many cosmeceuticals are green, since they have plant or microbial origin. Please, rephrase.

The explanation of “Biomolecule obtaining” is not a perspective. This part is not suitable here.

This item is wordy and does not bring real perspectives.

  1. Conclusions.

Also repetitive, should be combined with item 7, leaving only the conclusions, without repeating what was already said in the manuscript.

Author Response

Dear Reviewer,

We appreciate all your observations, we share the updated manuscript considering the comments of all reviewers. 

Regards

Reviewer 2 Report

General comment:

The review article “Microalgae bioactive compounds to produce microalgal-derived products for topical applications” - by M. Martínez-Ruiz and colleagues - reports the state of the art in several cosmeceutical categories where microalgae-derived biomolecules could be implemented. It considers microalgae bioactive molecules and their mechanism of action, applications, and results of implement in cosmeceutical products. The area of “topical application” for microalgal derived products is surely missing a proper state of the art at the moment, and within this context the paper is of great interest. However, due to the vast literature on the topic and the recent advances in microalgal biotechnology, I feel that the article still needs some improvements to fully meet the standards of the journal and interest of the readership.  In particular, issues as risk assessment and safety for human health cannot be neglected in the final paragraph Perspectives.
The article is overall well written, with the only exception of some typos (e.g. aditional VS additional in the abstract section) here and there in the text. Elsewhere in the manuscript some sentences could be shortened or made clearer with further attentive reading by the authors. I avoided punctual notifications for language because of missing line numbers. For the same reason, specific comments follow after a quotation from the text.

Regarding the general organization of the manuscript, I recommend:

  1. to standardize the structure of the paragraphs and numbering, i.e. adding coherent subsections in paragraphs 4-5-6 as in paragraphs 2 and 3 OR avoiding the use of subsections in paragraphs 2 and 3;
  2. to specify the taxonomic groups of the species mentioned, at least in the tables [i.e. Tetreaselmis (Chlorophyta); Nostoc (Cyanobacteria)];
  3. to specify the chemical description for the mentioned compounds of interest within the context (e.g. astaxanthin (pigment? Tetraterpenoid? Carotenoid? Xanthophyll-like or carotene-like?)

Specific comments

1)A: ”Because cosmeceuticals are compounds that have drug-like benefits while being a cosmetic product, they don’t have to be regulated by the U.S. Food and Drug Administration and they are not needed to prove safety or efficacy, as the term is still not recognized by this institution [4,5] Cosmeceuticals are mainly lotions or creams for topical use and aim to skincare, its importance is underestimated, although it’s calculated that 30% to 40% of dermatologist’s prescriptions around the world include a cosmeceutical [4]. “

R:
The authors state that cosmeceutical products are not regulated by the FDA but that determinologists' prescriptions include cosmeceuticals. Clarify the statement reported with respect to the FDA regulation and including other worldwide relevant regulations for these products (e.g. EU, China, Australia, Japan ...), and in light of this explain if the abovementioned prescriptions refer to prescription drugs or to over-the-counter products. Also, here the authors can provide few examples of microalgae derived cosmeceuticals which are available in the market (imaginative example: “anti-aging face cream with Dunaliella extracts”, “body sunscreen with Tetraselmis powder”, “regenerative mud with Spirulina”, etc ).

2) A: “Microalgae are a diverse group of photosynthetic eukaryotic microorganisms in various sizes, structures and forms, these organisms can produce valuable metabolites as a result of being constantly exposed to several stress conditions such as high or low temperature, high salinity, osmotic pressure, photo-oxidation, and ultraviolet radiation [7]. “

R:
Please rephrase the statement and clarify:

a) microalgae can be both eukaryotic and prokaryotic; the authors further on in the text report several species of cyanobacteria, which are not eukaryotes.

b) furthermore, the statement reported is only partially true. Microalgae in nature respond to environmental fluctuations with different strategies; however these fluctuations are not constant and permanent, and are not found in conditions of controlled massive cultivation intended for biotechnological purposes. Please give more context.

A: “They can grow in variate media such as wastewater and leachates as a result of the bioremediation process. “

R:
Please provide reference. Also, explain if possible that algae intended for wastewater treatment and bioremediation can be used in cosmeceutical applications. Does the accumulation of toxic substances and heavy metals not cause harm for human topical use?

4)A: “Levasseur et al. showed that in the last decades microalgae culture has been oriented to the production of high value-added molecules rather than environmental applications. It has been suggested that the commercial viability of microalgae production will be possible only if high added value molecules can be obtained from microalgae [14]. “

R:
Add reference [13] (Levasseur et al.) appropriately in the text.

5) A: “Some of the antioxidants which can be obtained from microalgae are chlorophyll, carotenoids, flavonoids and vitamins [13].“

R:
Please cite some opportune examples, with short reference to the function in the microalgal cell and differences from higher plants. All these classes of compounds are normally found in plants, mosses, and eve in lichens, what’s so special in microalgae? The section could be improved with following literature relevant to the topic:

https://doi.org/ 10.3390/md19070354

https://doi.org/10.1080/07388551.2021.1874284

https://doi.org/10.1186/s12934-020-01459-1

6) A:”Carotenoids have been identified in green microalgae like Spirogyra neglecta and Microspora indica [80]. “

R:
Please note that carotenoids are present in all algae divisions, even if in different proportions and/or with peculiar types; clarify the concept in the light of the cited bibliography. Also, the species names must be written in italics.

7) tables

R:

Please standardize Tables and elsewhere in the text:

a) specify which molecules are the cited "polysaccharides" "amino acides" or "flavonoids" ; e.g. in table 1 the compound "chrysolaminarin" is a polysaccharide indeed.

b) alternatively, standardize using a broader nomenclature for that type of compound, and replace, for example: - chlorophylls, carotenoids, astaxanthin – can be replaced with "pigments". C) please provide the division (phylum) of the various types of algae, at least in the tables. Being a heterogeneous and highly polyphyletic group, algae may be show great evolutionary distances among groups.

8) A:”Microalgae can produce a large variety of vitamins, including, vitamin A, B1, B2, B6, B12, C and E. Minerals can also be obtained from microalgae, like potassium, iron, magnesium, calcium and iodine [57,81]. Dunaliella tertiolecta is known to synthesize vitamin B12, B2, E and beta carotene [82]. Tetraselmis suecica produces vitamin C, which acts as a scavenger of reactive oxygen species and protects against lipid peroxidation via the reduction of lipid hydroperoxyl radicals [83].”

R:

Bibliography can be implemented in this section and further on in the text: for example, on the subject there are the following recently published works, which deal with the presence of these compounds in different algae divisions and particularly with biotechnological and even cosmeceutical applications.

https://doi.org/ 10.3390/antiox10091360
https://doi.org/10.1038/srep41215

https://doi.org/ 10.3390/md19070354
https://doi.org/10.1080/07388551.2021.1874284

https://doi.org/10.1186/s12934-020-01459-1

https://doi.org/10.3390/md18090477

9) A:”Ectoine is a compound that is highly common in halotolerant and halophilic microorganisms, its function is to protect the cells from the osmotic pressure due to the environments with high salt content, besides the osmolyte function [88],”

R:
Please, specify what kind of compound it is (from a chemical point of view) and if it falls in previously mentioned classes of compounds.

10) A: “Extracellular polysaccharides (EPSs) are polysaccharides that are secreted outside the cells by various marine organisms, including microalgae. These excretions represent a physical barrier protecting cells from harmful agents and environmental constraints and serve in a different physiological process including cell adhesion, interactions between cells and biofilm formation. The potential area of application for EPSs produced by microalgae and cyanobacteria includes industrial, pharmaceutical, and medical applications [139]. Depending on the concentration, they form viscous solutions to gels. Therefore, they can be used as active texturizing agents, carriers, or active protectors and enhancers [140]. Besides their application as gelling agents, EPSs are also important as emulsion stabilizers, flocculants, foam stabilizers and as hydrating agents (in cosmetics and pharmaceuticals) [141]. “

R:
Due to the aims of the paper, the description of the properties and applications of EPS deserves further study and an appropriate bibliography in the section; consider profound rewriting this passage, also providing one or two case-studies, due to the relevance of the topic in the context.

11) A:”Nanoalgosomes are novel membranous biogenic nanomaterial refined for the first time from a sustainable and renewable bioresource (microalgae), which can be used as a new natural delivery system for high-value microalgal substances (such as antioxidants, pigments, lipids and complex carbohydrates), bioactive biological molecules and/or synthetic drugs. “

R:
Please add reference for nanoalgosomes.

12) A:”Research has been made regarding the potential of porous silica-based particles for drug delivery applications due to their extended drug release profiles and high efficacy in delivering hydrophobic drugs [144]. “

R:

The feature here reported is typical only of diatoms among all microalgae. Specify this aspect with reference to classification. Currently there is large production on drug delivery applications derived from diatoms and properties of silica shell.

13) A:“Microalgae arises as a green alternative to cosmeceutical products.“

R:
This statement that opens the paragraph Perspectives summarizes some perplexities that the reader may have further on reading.

Land plants are also “green”, and the biotechnological yield alone does not seem to be convincing enough for the market to completely turn to algal production, whose limits are evident for the yield of pure molecules present in small intracellular concentrations.

Along the article the authors specifically refer to applications for topical use, but how should it be intended? Algal powder biomass as it is? Hot springs mud as it is? Extraction / fractioning / processing of the biomass? Different others or more than one among the previous? In all cases, please clarify the aspects related to “green chemistry” “renewable resources” “eco-friendly and sustainable production” which can be specific for the purpose of the readers interested in topical application of microalgae derived cosmeceuticals.
I feel that pipeline production of microalgae described in the section is beyond the scope of this paper, and - in my opinion - it could be surely shortened, if not completely deleted. On the other hand, safety for topical use in humans, market data, compliance of customers, types of cosmeceuticals based on microalgae, would be arguments of greater interest for the readership of this review. As a matter of fact, many species reported in the paper are able to secrete poisonous or even deadly toxins, while some other are totally safe and authorized for direct human consumption by national authorities, also including FDA. Some issues as risk assessment and safety for human health cannot be neglected in this final paragraph.

Author Response

(The authors gave the same response as above.)

Reviewer 3 Report

The manuscript ID molecules-1603511 entitled “Microalgae bioactive compounds to produce microalgal-derived products for topical applications” is an interesting and valuable study. The Authors carefully analysed the available literature summarized the state of the art in every cosmeceutical category where microalgae-derived biomolecules could be implemented, considering microalgae bioactive molecule mechanism of action, applications, and results of implement in cosmeceutical products. The manuscript is well written and the layout is correct and logical. The authors have actually separated the chapters. In my opinion, this is a fairly well-read study. It fully corresponds to the profile of the Molecules journal. The Authors properly present the current scientific achievements in the field of the microalgae bioactive compounds to produce derived products for topical applications.

My suggestions below:

  1. In my opinion, the title of the manuscript should be changed and simplified. The authors should consider the following version: "Microalgae bioactive compounds to topical applications products - a review".
  2. A graphical abstract would be very useful for the promotion of the manuscript and the understanding of the authors' intentions by the readers. Figure 1 is useful for this purpose.
  3. In my opinion, the Introduction provides not enough good background to this overview manuscript. In my opinion the use of microalgae to produce topical applications products is also inextricably linked with the concept of algal biorefinery as a sustainable approach to valorise algal-based biomass towards multiple product recovery. I suggest mentioning this issue in the introduction, because the biomass of microalgae after the recovery of valuable topical application products is used for fertilization, fodder or energy purposes: https://doi.org/10.3390/su12239980, https://doi. org / 10.1016 / j.biortech.2019.01.104. It is important because the multi-directional use of biomass improves the economics of the process.
  4. Please describe, present and evaluate the strengths and weaknesses production of the characterized products. An interesting issue would be the evaluation of technological and economic competitiveness of such solutions.

Author Response

(The authors gave the same response as above.)

Reviewer 4 Report

Authors are suggested following changes

Their are many English grammar mistakes so authors are advised to correct the English also representation of chemical names with sub scripts for eg  CO2 has to be corrected  for eg see section 6.

The animations in figure 1 and 2 are quiet similar, figures need to be made as per higher standard both in terms of their display as well their representation.

Authors are advised to expand the concept in section 3 "UV-A damages DNA indirectly, by generating radical singlet oxygen (1O2), resulting in purine base modifications, while UV-B directly damages DNA strands, generating pyrimidine dimers "

elaborate further section 3.2 and add one more section of ketocarotenoids about king of carotenoid molecules i.e astaxanthin and add biomolecules from red and brown algae and about astaxanthin and cite the following studies in their article:

In section 3 and 6 antioxidants add the following important studies

  1. Algae as sustainable food in space missions. In Biomass, Biofuels, Biochemicals (pp. 517-540).
  2. Bioprospecting of fucoxanthin from diatoms—Challenges and perspectives. Algal Research60, p.102475.

In section 6 add

"Exopolysaccharides directed embellishment of diatoms triggered on plastics and other marine litter. Scientific reports. 2020 Oct 28;10(1):1-1"

Conclusion should also focus on the technoeconomic importance of the molecules from microalgae, it needs reframing and formatting 

avoid old references like 2002 etc.

Author Response

(The authors gave the same response as above.)

Round 2

Reviewer 1 Report

The revised version of the manuscript shows some improvements and several comments were addressed. However, although I like the proposal, I still find that the theme was not well explored. 

Although leaving a record of the modifications is a request, the manuscript was so deeply modified that is difficult to read it now. I suggest to find a middle-term between showing the modifications and presenting a readable manuscript. It took me a long time to follow some points. 

The authors did not use the template correctly, therefore, again, it is not easy to point the page lines. In this way, I will give some general examples of weak points of the review.

Drawing the chemical structures did not solve the problem of figure 1. The structures inside the circular graph are still very small. The figure does not work. Anyway, the chemical structures were not correctly drawn. A scientific work must follow standards and the chemical structures have different formats.

The text was modified in several parts, however, a review paper must contain a coherent contribution of the authors, which I did not find.  In many points, there are only information of different papers, but I ask: what links are among the different papers? For example, the authors say "Additionally, methods of extraction and purification of the compounds of interest are investigated for their safe application in cosmetic and nutraceutical products". Ok, but which are these methods? What make them different? What is the connection with safety? Here and in other points, we expect the authors to give this kind of information.

Some new references were introduced in the reference list, but the manuscript still doesn't reflect the current literature, the advances in the area. 

Tables: I could not understand the correspondence between the information and the references cited in the table. When more than one reference is cited in the same line, I could not understand the relationship among them. Do they refer to different data?See, for example, Table 2 (Pigments). Why 3 references are cited? Did you get different information from different papers? It is confusing.

Again in Table 2. ROS absorption, elimination, reduction... Are these the same activity? If they are the same, why different terms? If they are different, please, explain.  The literature can present the same data using different terms. Here the authors give their contribution, choosing a standard term, so the reader can understand that the data can be comparable. Otherwise, if these different terms refer to different assays/mechanisms of action, it should be explained. I cited ROS, but it can be applied in other situations.

At the end, this manuscript is not ready yet in my opinion. It still need a recent review of the literature, with fresh data; the examples should be discussed, the innovation in the area must be highlighted. 

Author Response

REVIEWER 1

  • The revised version of the manuscript shows some improvements and several comments were addressed. However, although I like the proposal, I still find that the theme was not well explored. 

Authors Response:
The manuscript was updated according to the comments that you provide us to improve the manuscript.

  • Although leaving a record of the modifications is a request, the manuscript was so deeply modified that is difficult to read it now. I suggest to find a middle-term between showing the modifications and presenting a readable manuscript. It took me a long time to follow some points. 

Authors Response:
You can find in the attached manuscript, modifications in Tables 1, 2, 3, and 4, also in Figure 1. Each section was modified and improved traying to cover all the gaps noticed by the reviewers in redaction, grammar, and the full content.

  • The authors did not use the template correctly, therefore, again, it is not easy to point the page lines. In this way, I will give some general examples of weak points of the review.

Authors Response: 
The updated document is in PDF to avoid changes in the format from the template, containing numbered page lines, and the page number.

  • Drawing the chemical structures did not solve the problem of figure 1. The structures inside the circular graph are still very small. The figure does not work. Anyway, the chemical structures were not correctly drawn. A scientific work must follow standards and the chemical structures have different formats.

Authors Response:
Figure 1 was modified and standardized in the same format for chemical structures, also some graphical objects were removed to make clear the illustrations.

  • The text was modified in several parts, however, a review paper must contain a coherent contribution of the authors, which I did not find.  In many points, there are only information of different papers, but I ask: what links are among the different papers? For example, the authors say "Additionally, methods of extraction and purification of the compounds of interest are investigated for their safe application in cosmetic and nutraceutical products". Ok, but which are these methods? What make them different? What is the connection with safety? Here and in other points, we expect the authors to give this kind of information.

Authors Response:
The manuscript was modified in each section to make a coherent linking between cited papers, also at the end of each section, partial conclusions were included.

  • Some new references were introduced in the reference list, but the manuscript still doesn't reflect the current literature, the advances in the area. 

Authors Response:
As mentioned before, some sections were updated, and some of them were almost redone trying to show the current state of production, uses and advantages of microalgae extracts, and the perspective to improve those processes.

  • Tables: I could not understand the correspondence between the information and the references cited in the table. When more than one reference is cited in the same line, I could not understand the relationship among them. Do they refer to different data?See, for example, Table 2 (Pigments). Why 3 references are cited? Did you get different information from different papers? It is confusing.
    Again in Table 2. ROS absorption, elimination, reduction... Are these the same activity? If they are the same, why different terms? If they are different, please, explain.  The literature can present the same data using different terms. Here the authors give their contribution, choosing a standard term, so the reader can understand that the data can be comparable. Otherwise, if these different terms refer to different assays/mechanisms of action, it should be explained. I cited ROS, but it can be applied in other situations.

Authors Response:
All tables were modified and references were separated to avoid confusion. ROS scavenger and attenuation were used as a proper description of the mechanism of action.

Reviewer 2 Report

General comment

Although the work has improved from some points of view, especially the formal ones, on the other hand it remains poor as regards the critical analysis of some points that are fundamental to the theme that the authors have decided to deal with.
In particular, some "hot spots" that could be critically discussed and give greater impact and novelty to the work, were omitted or ignored.
I am referring - for example, but not only - to the regulation regulations for cosmeceutical products, the types of formulations that can be obtained, the extraction of the molecules of interest and the fact that microalgae constitute a "green" alternative, problems that authors raise, without however going deeper, or even misleading the reader.
I realize that the review process takes into account the suggestions of the different reviewers, but at the same time the perplexity shared by the reviewers about the bibliography does not seem to have been solved. Indeed, entire periods inserted in this new version of the work lack appropriate references: in the version that I can see, the numbering of the lines is partial, so once again I cannot make precise references to the text, but I refer for example to the cost per gram of fucoxanthin, or to the section of perspectives that entirely lacks bibliographic references, resulting practically anecdotal.
Being a review article, the reader should be able to easily trace the sources relevant to the intended purposes. In this context, the suggestion to give examples of already available microalgae-based cosmeceuticals was also overlooked.
Finally, it is my feeling that the revision took place quite hastily, since the edited text shows that language is poor or even incorrect.

Specific comments

The new figure 1 shows an ambiguous and misleading caption; while laminarine and chrysolaminarine are polysaccharides (and are rightly polymeric) astaxanthin and lycopene cannot be considered polymers (and in fact they are pigments, carotenoids). Furthermore, the figure does not correspond to the description in the text, in which several types of compounds are mentioned.

Table 4 should be modified like the previous ones.

Author Response

REVIEWER 2

General comment
Although the work has improved from some points of view, especially the formal ones, on the other hand it remains poor as regards the critical analysis of some points that are fundamental to the theme that the authors have decided to deal with. In particular, some "hot spots" that could be critically discussed and give greater impact and novelty to the work, were omitted or ignored. I am referring - for example, but not only - to the regulation regulations for cosmeceutical products, the types of formulations that can be obtained, the extraction of the molecules of interest and the fact that microalgae constitute a "green" alternative, problems that authors raise, without however going deeper, or even misleading the reader.

Authors Response:
The manuscript was updated according to the comments that reviewers provide us to improve the manuscript. You can find in the attached manuscript modifications in Tables 1, 2, 3, and 4, also in Figure 1. Each section was modified and improved traying to cover all the gaps noticed by the reviewers in redaction, grammar, and the full content.

I realize that the review process takes into account the suggestions of the different reviewers, but at the same time the perplexity shared by the reviewers about the bibliography does not seem to have been solved. Indeed, entire periods inserted in this new version of the work lack appropriate references: in the version that I can see, the numbering of the lines is partial, so once again I cannot make precise references to the text, but I refer for example to the cost per gram of fucoxanthin, or to the section of perspectives that entirely lacks bibliographic references, resulting practically anecdotal.

Authors Response:
The updated document is in PDF to avoid changes in the template format. The document contains numbered page lines, and page numbers for proper reference in your comments. you can find some improvements in all sections.

Being a review article, the reader should be able to easily trace the sources relevant to the intended purposes. In this context, the suggestion to give examples of already available microalgae-based cosmeceuticals was also overlooked.
Finally, it is my feeling that the revision took place quite hastily, since the edited text shows that language is poor or even incorrect.
Authors Response:
The manuscript corrections attend language improvement to trying express the right message in each section. 

Specific comments
The new figure 1 shows an ambiguous and misleading caption; while laminarine and chrysolaminarine are polysaccharides (and are rightly polymeric) astaxanthin and lycopene cannot be considered polymers (and in fact they are pigments, carotenoids). Furthermore, the figure does not correspond to the description in the text, in which several types of compounds are mentioned.
Authors Response:
Figure 1 was modified and standardized the structures of the molecules according to the different application areas.

Table 4 should be modified like the previous ones.
Authors Response:
As mentioned before, all tables were improved to avoid confusion about references and content.

Reviewer 3 Report

Thanks Author's for improving manuscript. In my opinion it can be publish in current form.

Author Response

REVIEWER 3

Thanks Author's for improving manuscript. In my opinion it can be publish in current form.

Authors response: 

You can find in attach the improved version according to the other reviewers comments.

Reviewer 4 Report

Authors have satisfactorily done the revision

Author Response

REVIEWER 4

Authors have satisfactorily done the revision

Authors response: 

You can find in attach the improved version according to the other reviewers comments.
